# Low-Energy Shock Wave Plus Intravesical Instillation of Botulinum Toxin A for Interstitial Cystitis/Bladder Pain Syndrome: Pathophysiology and Preliminary Result of a Novel Minimally Invasive Treatment

**DOI:** 10.3390/biomedicines10020396

**Published:** 2022-02-07

**Authors:** Yuan-Hong Jiang, Jia-Fong Jhang, Yu-Khun Lee, Hann-Chorng Kuo

**Affiliations:** Department of Urology, Hualien Tzu Chi Hospital, Buddhist Tzu Chi Medical Foundation, Buddhist Tzu Chi University, Hualien 97004, Taiwan; redeemer1019@yahoo.com.tw (Y.-H.J.); alur1984@hotmail.com (J.-F.J.); leeyukhun@gmail.com (Y.-K.L.)

**Keywords:** low-energy shock wave, interstitial cystitis, botulinum toxin A, urothelium

## Abstract

Low-energy shock wave (LESW) therapy is known to facilitate tissue regeneration with analgesic and anti-inflammatory effects. LESW treatment has been demonstrated to be effective in treating chronic prostatitis and pelvic pain syndrome as well as overactive bladder, and it has a potential effect on interstitial cystitis/bladder pain syndrome (IC/BPS) in humans. LESW reduces pain behavior, downregulates nerve growth factor expression, and suppresses bladder overactivity by decreasing the expression of inflammatory proteins. Previous rat IC models have shown that LESW can increase urothelial permeability, facilitate intravesical delivery of botulinum toxin A (BoNT-A), and block acetic acid-induced hyperactive bladder, suggesting that LESW might be a potential therapeutic module for relieving bladder inflammatory conditions, such as bladder oversensitivity, IC/BPS, and overactive bladder. A recent clinical trial showed that LESW monotherapy was associated with a significant reduction in pain scores and IC symptoms. BoNT-A detrusor injection or liposome-encapsulated BoNT-A instillation could also inhibit inflammation and improve IC symptoms. However, BoNT-A injection requires anesthesia and certain complications might occur. Our preliminary study using LESW plus intravesical BoNT-A instillation every week demonstrated an improvement in global response assessment without any adverse events. Moreover, an immunohistochemistry study revealed the presence of cleaved SNAP25 protein in the suburothelium of IC bladder tissue, indicating that BoNT-A could penetrate across the urothelial barrier after application of LESW. These results provide evidence for the efficacy and safety of this novel IC/BPS treatment by LESW plus BoNT-A instillation, without anesthesia, and no bladder injection. This article reviews the current evidence on LESW and LESW plus intravesical therapeutic agents on bladder disorders and the pathophysiology and pharmacological mechanism of this novel, minimally invasive treatment model for IC/BPS.

## 1. Introduction

Interstitial cystitis/bladder pain syndrome (IC/BPS) is a chronic bladder condition characterized by bladder pain, frequency, and nocturia [1]. Bladder and pelvic pain, glomerulations after cystoscopic hydrodistention, and denudation of the bladder urothelium are the most common clinical characteristics of IC/BPS, suggesting that bladder inflammation and urothelial dysfunction are the key contributing factors [1,2]. However, the actual pathophysiology of IC/BPS remains unclear. The apoptotic process in the microvascular endothelial cells and high levels of urothelial cell apoptosis have been detected in IC/BPS bladders [3,4]. As the pathogenesis of IC/BPS has not been sufficiently elucidated, there is no definite treatment that provides long-term symptom relief. Recently, efforts have been made to treat IC/BPS by improving urothelial cell regeneration and ameliorating chronic inflammation. This article reviews recent research into the pathophysiology of IC/BPS and the potential therapeutic effects of low-energy shock wave (LESW) for treating IC/BPS.

## 2. Chronic Inflammation and Urothelial Dysfunction Are the Fundamental Pathophysiology of IC/BPS

A previous study revealed significant impairment in urothelial homeostasis in IC/BPS bladders and found that a deficient urothelial barrier was associated with chronic inflammation and increased cell apoptosis of the urothelium [4]. Another study found that, with respect to overactive bladder (OAB), chronic inflammation was found in approximately 60% of biopsies of the bladder urothelium [5]. The expression of inflammatory markers, such as in mast cells in IC/BPS bladders, is similar to that in OAB [6]. Urinary inflammatory biomarkers, such as nerve growth factor (NGF), eotaxin, CXCL10, and RANTES, were also noted to be elevated in patients with OAB as well as with IC/BPS [7,8]. An immunohistochemistry study demonstrated that the pathogenesis of IC/BPS involves an aberrant differentiation program in the urothelium, leading to the altered synthesis of mucosal surface proteoglycans, cell adhesion proteins, tight junction proteins, and bacterial defense molecules, such as GP51 [9]. IC/BPS may originate from an inflammatory process that alters the regulation of urothelial homeostasis and results in dysfunction of the bladder epithelium. In IC/BPS bladders, chronic inflammation, increased urothelial apoptosis, and abnormal urothelial barrier function are closely associated with each other [10].

Excessive levels of one or more of the inflammatory cytokines, such as tumor necrosis factor (TNF)-α, interleukin (IL)-6, or IL-8, are usually found in chronic inflammatory diseases [11]. Our recent study showed that the apoptotic signaling molecules, including Bad, Bax, and cleaved caspase-3, were increased in IC/BPS bladders [12]. These results suggest that apoptosis in the IC/BPS bladder urothelium might be due to an upregulation of inflammatory signals.

The proliferation and differentiation of the stem cells and progenitor cells in the basal cell layer regulate epithelium integrity in the urinary bladder. Urothelial cells undergo rapid proliferation during urothelial injury with chemicals, toxins, or bacterial infection [13]. The urothelial basal cells are considered to be progenital cells for epithelial repair, which is marked by the expression of the secreted protein Sonic Hedgehog (Shh). The Shh expression in the basal cells increases upon urothelial injury, followed by the stromal expression of *Wnt* protein signals, increasing and stimulating the proliferation of urothelial and stromal cells [13]. The histopathology of IC/BPS includes mast cell infiltration, suggesting that the pathogenesis of IC/BPS is mediated by an abnormal immune system and chronic inflammation [2,14]. Recently, we also found that the bladder urothelium of Hunner’s ulcer IC/BPS had a higher grade of moderate or severe eosinophil infiltration and urothelium denudation than that in non-ulcer IC/BPS bladders [15]. In comparison with non-ulcerative IC/BPS, the adhesive protein E-cadherin expression in the urothelium was significantly lower, and endothelial nitric oxide (NO) synthase expression was significantly higher in Hunner’s IC/BPS bladder [16]. Using the cell markers CK5, CK14, CK20, and Shh to investigate the regenerative ability of urothelial cells in IC/BPs bladders, we found that Hunner’s IC/BPS bladders had a low proliferative and differentiation ability, whereas the other non-ulcerative IC/BPS bladders also had a deficit of proliferative and differentiation ability, both of which were also associated with the grade of glomerulations under hydrodistention [15] (Figure 1).

## 3. Inflammation-Induced Oxidative Stress Inhibits Normal Urothelial Proliferation and Differentiation in IC/BPS

Patients with IC/BPS had urine inflammatory cytokine profiles that were different from those in the controls. The urine cytokine profiles were also different between ESSIC type 1 and type 2 IC/BPS patients [17]. Chronic inflammation in the IC/BPS bladders might impair the proliferative ability of the basal cells, alter the differentiation rate, and result in defective mature urothelial apical cells and barrier function [18]. To restore normal bladder mucosal barrier function, a treatment that aims to improve the proliferation of basal cells in diseased bladders, such as IC/BPS, is important.

Besides the inflammatory cytokines, urine oxidative stress might also contribute to the pathophysiology of IC/BPS, and oxidative stress biomarkers have the potential to be used as a diagnostic tool for IC/BPS. Previous studies have shown that diminished bladder perfusion occurred in IC/BPS. Hypoxia induces the overexpression of vascular endothelial growth factor (VEGF), which might be associated with glomerulation in the IC/BPS bladders after hydrodistention. Immunoblotting and immunostaining of IC/BPS bladders revealed that the expression of hypoxia-inducible factor-1 alpha (HIF-1α) and VEGF proteins was increased in IC/BPS bladders when compared with the controls [19]. Tissue hypoxia and hypoxia-induced signaling pathways might play crucial roles in disease progression and bladder remodeling of bladder outlet obstruction [20,21]. Thus, in IC/BPS bladders, treatment targeting the reversal of bladder perfusion and a reduction in bladder inflammation might reduce bladder pain, increase functional bladder capacity, and reduce glomerulation grade [22].

Several urine oxidative stress biomarkers have the potential to serve as novel biomarkers in IC/BPS, including 8-OHdG [23], F2-isoprostane [24], and total antioxidant capacity (TAC), and they reflect the cumulative effect of all antioxidants from various endogenous antioxidative defense systems against harmful activities caused by oxidative stress [25]. One recent review disclosed that 8-OHdG, F2-isoprostane, and TAC were applied as oxidative stress and antioxidant biomarkers in urinary dysfunction related to bladder outlet obstruction [26]. Recently, we investigated the roles of oxidative stress biomarkers in IC/BPS patients. Both 8-OHdG and 8-isoprostane were found to be independent analytes in urine that could be used to discriminate IC/BPS from the controls. In urine, their levels even provided a higher area under the curve than selected inflammatory cytokines for discriminating ESSIC type 2 IC/BPS from controls [8]. Additionally, significant correlations between urine oxidative stress biomarkers and clinical characteristics of ESSIC type 2 IC/BPS patients were demonstrated.

## 4. Deficits of Urothelial Barrier Result in Bladder Pain Syndrome

Urothelial dysfunction has been widely accepted as the main pathogenesis of IC/BPS and has become a therapeutic target [16]. Previous research indicated that deficiency of mucosal glycosaminoglycan, which causes an increase in urothelial permeability, is an important cause of painful bladder symptoms in IC/BPS patients [27,28]. An immunohistochemistry investigation also revealed that urothelial cellular tight junction zonula occludens-1 (ZO-1), and adhesive protein E-cadherin were downregulated in IC/BPS bladders [6]. Additionally, upregulation of the purinergic receptor P2X3 and an increase in the release of adenosine triphosphate (ATP) from the urothelium have been found in IC/BPS bladders [29,30]. Up to now, urothelium dysfunction has been considered one of the key pathogenetic and treatment targets of IC/BPS.

Electron microscopy (EM) studies have reported a deficiency of the cellular tight junction, epithelial cell pleomorphism, microvilli of the urothelial cell membrane, and mast cells in IC/BPS bladders [31,32]. In our recent EM study, we also found a significant decrease in cell layers and a decrease in apical umbrella cell integrity in IC/BPS urothelium [33]. Compared with control bladders, IC/BPS urothelium had a significantly more severely deficient urothelial cell layer and loss of integrity of apical umbrella cells, as shown in transmission EM. Meanwhile, the increase in umbrella cell pleomorphism and the decrease of microplicae of the apical cell membrane were remarkable in the scanning EM of IC/BPS bladders (Figure 2). Both EM findings showed significantly greater defective features in IC/BPS bladders than in the controls. The greater deficiency in the urothelial cells and impaired umbrella cell integrity shown in EM were associated with a higher degree of bladder pain and smaller maximal bladder capacity [33].

These EM findings suggest that urothelial dysfunction might result from defective umbrella cell coverage and cause more severe bladder pain scores in patients with IC/BPS. The loss of apical umbrella cells of the urothelium results in a deficit in barrier function, leading to an influx of urinary solutes or ionic substances across the defective urothelium to the suburothelial tissue and directly eliciting bladder pain in patients with IC/BPS [34]. The microplicae or ridges in the umbrella cell membrane usually become flattened during bladder distension and may play an important role in normal bladder physiology [35]. A decrease in microplicae of the apical cell membrane is considered to be associated with immature apical cells (actually, the intermediate cells but not umbrella cells) and may restrict the extent of bladder distention, resulting in small functional bladder capacity and frequency symptoms. Additionally, apical cell pleomorphism was noted on scanning EM, suggesting the partial loss of umbrella cells, which were replaced by immature intermediate cells (Figure 2). The findings of the EM study suggest that the deficiency in the urothelium, especially in the loss of mature umbrella cells, may play an important role in the pathogenesis of IC/BPS [36]. When compared with control and NHIC groups, Hunner’s IC/BPS patients exhibited more severe defects in the urothelium cell layers, including a greater loss of umbrella cells, umbrella cell surface uroplakin plaque, and tight junctions between adjacent umbrella cells. The defects in the urothelium cell layer on EM were associated with greater severity of clinical symptoms.

## 5. Intravesical BoNT-A Injection, Liposome-Encapsulated BoNT-A, and Platelet-Rich Plasma Injections Decrease IC/BPS Symptoms

### 5.1. Intravesical BoNT-A Injection

Botulinum toxin A (BoNT-A) is a potent neurotoxin produced by the bacterium *Clostridium botulinum* [37]. When the toxin enters the cell through endocytosis, it is cleaved into a heavy chain (100 kDa) and a light chain (50 kDa) by proteolytic cleavage, and the toxin becomes biologically active [38]. BoNT-A molecules enter the neuronal cell membrane by binding to the synaptic vesicle protein SV2. Through endocytosis, the toxin is cleaved into heavy and light chains [39]. The light chain of BoNT-A binds to the SNAP-25 protein and inhibits the neurotransmitters released from the vesicles [40]. BoNT-A can inhibit neurotransmitters and neuropeptides, including acetylcholine (Ach), ATP, NO, substance P, and calcitonin gene-related peptide (CGRP) [41]. BoNT-A injection can relax striated or smooth muscles with neurotransmitters of Ach and ATP and also controls local inflammation mediated via substance P and CGRP [42]. BoNT-A can also block transient receptor potential vanilloid subfamily-1 (TRPV1) and purinergic receptor P2X3-IR expressions and decrease bladder sensory disorders by desensitizing TRPV1 and P2X3 [43]. Hence, intravesical BoNT-A injection can reduce bladder oversensation and bladder pain as well. It is also postulated that BoNT-A might control chronic neurogenic pain by acting on the peripheral nociceptive neurons and central desensitization by retrograde transport of BoNT-A to the central nervous system [44,45].

Recent research on the pathophysiology of IC/BPS has found that the increase in urothelial cell apoptosis was mediated by suburothelial inflammation of the bladder [4,12]. Intravesical BoNT-A injection can inhibit chronic inflammation, improve the barrier function of the bladder urothelium, and therefore reduce symptoms of patients with IC/BPS [46]. Neural upregulation in association with chronic inflammation has been considered to participate in the pathophysiology of IC/BPS [47]. Upregulation of the P2X3 receptor during the stretch of urothelial cells of the bladder had been found in patients with IC/BPS [29]. Additionally, increased mRNA expressions of the genes involved in pronociceptive inflammatory reactions in IC/BPS, including TRPV1, TRPV2 and TRPV4, ASIC1, NGF and CXCL9, and TRPM2, have been found in IC/BPS bladders [48]. These findings suggest that bladder inflammation might alter the neuropeptide expressions and increase bladder excitability and sensitization, resulting in increased pain sensation during bladder distention, thus causing frequency urgency and bladder pain in IC/BPS patients [49]. Since 2004, BoNT-A has been used for the treatment of IC/BPS. Our previous study revealed that BoNT-A could effectively relieve bladder pain in patients with IC/BPS refractory to conventional treatments [50]. Later, several clinical studies also proved the therapeutic efficacy of BoNT-A on IC/BPS [51,52] and significant improvement of anxiety, depression, and quality of life in patients with IC/BPS after BoNT-A injection [53].

In our prospective, randomized controlled trials in patients with IC/BPS refractory to treatment, we demonstrated that a suburothelial BoNT-A injection plus cystoscopic hydrodistention significantly improved visual analog scale (VAS) scores of bladder pain, functional and cystometric bladder capacity, and global response assessment at 3 months after treatment, when compared with patients receiving cystoscopic hydrodistention alone [46,54]. Repeat BoNT-A injections every 6 months increased the therapeutic duration, and the therapeutic duration was shown to be longer in patients receiving four consecutive BoNT-A treatments [55]. Nevertheless, patients who are scheduled to receive a BoNT-A injection should be informed of the potential adverse events, such as large postvoid residual (PVR) and acute urinary retention, and should be educated on the need for clean intermittent catheterization after BoNT-A injection [56,57]. A recent clinical trial demonstrated the therapeutic efficacy and safety of these injections. Intravesically suburothelial injections of 100 U BoNT-A significantly reduced bladder pain symptoms and increased cystometric bladder capacity when compared with individuals receiving normal saline. The overall success rate was 63% in the BoNT-A group and 15% in the normal saline group [58]. Although BoNT-A treatment for IC/BPS has not been approved by the regulatory authorities, this treatment has been documented in the IC/BPS clinical guidelines of the American Urological Association and Asian Urological Association [59,60].

### 5.2. Liposome-Encapsulated BoNT-A (Lipotoxin)

As traditional BoNT-A treatment for IC/BPS requires intravesical injections under local or general anesthesia, there has been greater enthusiasm for testing to determine whether intravesical instillation of BoNT-A with the aid of a vehicle is possible. Liposomes comprise phospholipids that can adhere to the cell membrane of apical urothelial cells; thus, liposomes have been used to counter mucosal inflammation and promote wound healing in IC/BPS [61]. Additionally, liposomes were found to carry drugs that penetrate across the barrier of the urothelial cell membrane. In a pilot study, we found that liquid liposomal delivery of BoNT-A (liposome—BoNT-A, Lipotoxin) could penetrate the bladder urothelium in patients with OAB refractory to antimuscarinic therapy [62]. After intravesical instillation of Lipotoxin (containing 80 mg of liposomes mixed with 200 U of BoNT-A) or normal saline, we found that Lipotoxin effectively reduced the frequency episodes and urgency severity score when compared with the normal saline treatment group in OAB patients, with adverse events including an increase in PVR and urinary tract infection (UTI) [62]. The SV2A receptors were present in human urothelial cell lysate. Nevertheless, soluble *N*-ethylmaleimide sensitive factor attachment protein receptor SNAP25 did not show a significant decrease in OAB bladder tissues [62]. Another multicenter clinical trial using Lipotoxin for OAB patients revealed a significant decrease in frequency urgency episodes in patients receiving Lipotoxin treatment, without increasing the PVR volume [63]. These clinical trial results suggest that Lipotoxin could block the release of sensory neurotransmitters in the urothelial cells, inhibit sensory hyperactivity, but not affect detrusor contractility. Although liposome-encapsulated BoNT-A to treat OAB or IC/BPS seems rational and promising, only 50% of OAB patients responded to this treatment, and the therapeutic effect on IC/BPS was not superior to that of placebo [64]. Nevertheless, in patients with moderate-to-severe IC/BPS, a single instillation of intravesical Lipotoxin was associated with decreased IC symptoms when compared with baseline [64]. Technical improvements in the formulation of liposomes, dose of BoNT-A, and instillation modality might increase the response rate in the future.

### 5.3. Platelet-Rich Plasma

Since current treatments for IC/BPS are usually unsuccessful in achieving long-term improvement, treatments that aim to ameliorate bladder inflammation, increase bladder perfusion, and promote urothelial regeneration have been attempted. Autologous platelet-rich plasma (PRP) is growing in popularity as a therapy to facilitate wound healing, speed the recovery of muscle and joint injuries, and enhance wound recovery after surgical repair [65]. PRP is rich in several growth factors and cytokines that modulate the inflammatory process and tissue regeneration in the wound healing process. An initial clinical trial demonstrated that intravesical PRP injection was safe and effective in reducing bladder pain and increasing the functional bladder capacity of IC/BPS bladders [66]. Repeated intravesical injections of autologous PRP were shown to increase bladder capacity and provide IC symptom improvement in patients with IC/BPS refractory to conventional therapy. In selected patients, autologous PRP injections are safe and effective [67]. After repeat PRP injections, urinary levels of NGF, matrix metalloproteinase-13, and VEGF decreased significantly, whereas platelet-derived growth factor-AB showed a significant increase on the fourth PRP treatment when compared with baseline [68]. Our previous immunohistochemistry study and EM study revealed an evident loss of normal umbrella cells and defective junction proteins in the bladder urothelium of IC/BPS and recurrent cystitis [69]. In the bladders of patients with recurrent UTI, repeat intravesical PRP injections improved the proliferation of urothelial cells and increased the expression of CK20 in umbrella cells. Hence, repeat PRP injections may restore urothelial health and prevent UTI recurrence in intractable recurrent UTIs and could have similar benefits in IC/BPS bladders [70].

## 6. Improvement of Urothelial Regeneration in IC/BPS Bladders after Successful Treatment

The bladder urothelium serves as a barrier to prevent injurious stimuli, toxins, or microorganisms from invading the stroma and upper urinary tract [71]. In the human bladder urothelium, apical umbrella cells, intermediate cells, and basal cells can be distinguished by cell membrane molecular markers. Mature apical cells express low-molecular-weight cytokeratin 20 (CK20) and uroplakins but not the high-molecular-weight cytokeratin CK5, transcription factor p63, or signaling molecule Shh. Intermediate cells of the urothelium express p63, Shh, and Upk but not Krt5, Krt14, or Krt20. Basal cells express Krt5, Shh, and p63, but not uroplakins or CK20 [13,72]. Approximately 14% of CK5-positive basal cells also express CK14, which are considered to be progenitor cells in urothelial regeneration. These combinations of urothelial-specific markers can be used to differentiate the proliferation, differentiation, and maturation of urothelial regeneration in different bladder disorders.

Urothelial dysfunction resulting in barrier defects, chronic inflammation, increased apoptosis of urothelial cells, nociceptive receptor upregulation, mast cell activation, and somatic functional syndrome constitutes the pathophysiology of IC/BPS [2,12]. Ultrastructural investigation revealed that the fundamental urothelial dysfunction of IC/BPS consists of the loss of mature apical umbrella cells and defects of the cellular tight junction and adhesive proteins [33]. With persistent suburothelial inflammation in IC/BPS bladders, increased apoptosis of urothelial cells and decreased cell proliferation result in defective mucosal integrity and increased urothelial permeability, causing bladder irritative symptoms [4]. The pathophysiology of IC/BPS might result from the impaired regenerative ability of the urothelial cells. If progenitor cell regeneration can be improved, the function of the bladder’s urothelial barrier might be rebuilt, and IC/BPS symptoms could also be eliminated.

A recently published clinical trial using four monthly PRP intravesical injections to treat patients with refractory IC/BPS demonstrated a high success rate [68]. Forty patients completed the four PRP injections and posttreatment visits. GRA improved after the first PRP treatment and persisted up to 3 months after the fourth PRP treatment. After PRP treatment, the functional bladder capacity, and maximum flow rate increased and the VAS for pain, IC symptoms, problem indexes, O’Leary–Sant symptom score, and GRA improved in all patients. Related urinary functional biomarkers also showed significant changes after repeat intravesical PRP injections in association with symptom improvement [68]. Expressions of urothelium *Shh,* CK5, ZO-1, E-cadherin, and transforming growth factor β also increased significantly after repeated PRP injections. By subgrouping according to PRP treatment outcomes, significant increases in *Shh*, E-cadherin, and ZO-1 expressions were noted only in patients with a GRA ≥ 1 or improved VAS but not in patients with a GRA of 0 and no improvement in VAS [73]. Besides the IC symptom scores, we could investigate the therapeutic efficacy of treatment for IC/BPS patients using the urothelial barrier markers, cytoskeletal maturation markers, and progenital cell markers (Figure 3).

## 7. LESW Reduces Tissue Inflammation and Attenuates Pain and Bladder Dysfunction in Urological Disorders

LESW is a kind of stress wave that carries energy and propagates through a medium to achieve desired biological effects [74]. A previous study revealed that the use of LESW in chronic injured tissue could reduce inflammation, promote neovascularization, and facilitate tissue regeneration [75]. LESW has now been widely used for treating orthopedic diseases, such as pseudoarthrosis, tendinitis, calcarea of the shoulder, epicondylitis, and plantar fasciitis through an anti-inflammatory effect, increase in neoangiogenesis, and tissue regeneration [75,76]. In the urinary bladder, a previous animal study in rats with cyclophosphamide-induced bladder inflammation showed that LESW could reduce bladder pain, urinary frequency, and inflammation [77].

Recent animal and human studies have shown that LESW can be applied to treat several different bladder disorders by using therapeutic effects to increase vasculature, reduce inflammation, and facilitate reinnervation. LESW also ameliorates bladder dysfunction and urinary continence in a rat model of diabetic bladder dysfunction, and the results showed that LESW can restore the nerve expression of the urethra and the vascularization of the bladder [78,79]. In a cryoinjury-induced detrusor underactivity rat model, LESW was shown to improve the composition of the bladder wall and hasten functional recovery [80]. In rats with hydrochloride-induced cystitis, LESW demonstrated the ability to suppress bladder overactivity by attenuating mitochondrial dysfunction through the reversal of the mitochondrial-dependent intrinsic apoptotic pathway [81]. In a UPK3A-induced IC/PBS rat model, LESW treatment attenuated pain and bladder dysfunction. LESW was used to raise the pain threshold and improve bladder function. Local and systemic inflammation biomarkers TNF-α and NGF were significantly reduced after LESW treatment [82].

LESW has also been used to treat male patients with prostatitis-like symptoms without remarkable adverse effects [83]. In a recent prospective, randomized clinical trial, patients with OAB symptoms who received LESW (0.25 mJ/mm^2^, 3000 pulses, three pulses/s) once a week for 8 weeks exhibited significant symptom improvement in comparison with sham controls, and the therapeutic effect lasted up to 6 months [84].

## 8. LESW Ameliorates Bladder Inflammation and Might Promote Urothelial Cell Regeneration in IC/BPS Bladders

Therapeutic approaches have been devoted to searching for a method that can deliver BoNT-A molecules across the barrier boundary of the bladder mucosa without the need for needle injections, such as using protamine sulfate to disrupt the barrier of the urothelium; electromotive drug delivery and LESW to increase the permeability of the urothelium; and liposomes, thermosensitive hydrogel, and hyaluronan–phosphatidylethanolamine to create a carrier for BoNT-A transportation [85]. In a rat model of bladder cancer, LESW applied on the bladder plus intravesical instillation of epirubicin was used to enhance the delivery of a chemotherapeutic agent into the mucosa. The combination of intravesical epirubicin and LESW results in reduced cancer cell invasion and urothelial cell dysplasia. In that study, LESW was shown to increase urothelial permeability and enhance the delivery of the drug into tumor tissues without subsequent toxicity [86].

LESW is also well known for its effect on altering cell membrane permeability and providing a noninvasive method for macromolecular delivery into cells [87,88]. Recently, a rat study showed that LESW could mediate the transdermal delivery of a local anesthetic drug into caudal nerves [89]. LESW has been shown to have an anti-inflammatory effect and to reduce pain in cyclophosphamide- or radiation-induced cystitis in rats. From the bladder protein expression of inflammatory and oxidative stress biomarkers, LESW seems to be effective in protecting the urinary bladder from cyclophosphamide- or radiation-induced cystitis in rats [77,90]. In musculoskeletal disorders and myofascial pain syndrome, LESW therapy can effectively increase blood circulation in the treatment area and reduce muscle hypertonicity and spasticity [91,92]. Furthermore, LESW can inhibit apoptosis, enhance cell proliferation, and improve wound healing by triggering the release of cellular ATP, which subsequently activates purinergic receptors and enhances proliferation via downstream Erk1/2 signaling [93]. This therapeutic effect might facilitate the regeneration of the wounded urothelium in the IC/BPS bladders. In a previous animal study showing the analgesic effect of LESW, rapid degeneration of the intracutaneous nerve fibers after LESW application to the skin was demonstrated, and reinnervation occurred approximately 2 weeks after treatment with LESW [94]. The reduction in bladder pain after LESW treatment might be due to the degeneration of C-afferent fiber.

Chuang et al. recently performed a prospective, randomized, and placebo-controlled study to evaluate the efficacy of LESW in patients with IC/BPS [95]. Patients with IC/BPS were randomly assigned to receive LESW applied at the suprapubic area once a week for 4 weeks or sham treatment. The results showed that in both treatment and control groups, the VAS pain score was statistically significantly reduced, but there was no significant difference in pain reduction between the two groups. Only two patients in the LESW group reported mild suprapubic pain during treatment. No patient in either of the groups experienced adverse events, such as urinary incontinence, urinary retention, or UTI. This study showed that LESW monotherapy in IC/BPS bladders is safe and has a therapeutic effect, especially in pain reduction.

## 9. LESW Enhances Penetration of BoNT-A across the Urothelium in IC/BPS Bladder: Preliminary Results

Recently, a study showed that LESW could increase urothelial permeability, facilitate intravesical BoNT-A delivery, and block acetic acid-induced hyperactive bladder in rats. These results provide support for the promising use of LESW to deliver the BoNT-A molecule across the barrier of the bladder urothelium, without the need for intravesical injections [96]. In that study, intravesical instillation of BoNT-A after LESW application was able to suppress the acetic acid-induced inflammatory reaction and hyperactivity of the bladder. Immunohistochemical staining also revealed decreases in the expression of SANP23 and SNAP-25 in rat bladders with intravesical instillation of BoNT-A after LESW. This evidence showed that suprapubic LESW treatment could promote the delivery of BoNT-A into the urothelium and may have the potential for clinical use [96]. Another OAB rat study using an intravesical instillation of BoNT-A plus LESW treatment showed a statistically significantly lower amplitude and lower frequency of detrusor contractions in the rats receiving treatment. LESW plus BoNT-A was also associated with statistically significantly lower expressions of TNF-α and IL-6 and significantly reduced submucosal edema and inflammatory cell infiltration [97]. LESW has been shown to be effective in increasing urothelial permeability to BoNT-A and in enhancing the therapeutic effect on the OAB rat model, possibly through anti-inflammatory and antioxidative stress effects.

One recent clinical trial in OAB patients also showed that LESW is a safe and effective method for treating refractory OAB with a response duration of 2 months [84], while intravesical instillation of BoNT-A plus LESW showed statistically significantly lower amplitude and lower frequency of detrusor contractions in the OAB rats [97].To explore the potential clinical efficacy of this treatment module in patients with IC/BPS, we previously performed a pilot clinical study of a case series using suprapubic LESW to promote intravesical BoNT-A delivery. At baseline, six women with refractory IC/BPS underwent cystoscopic hydrodistention and bladder biopsy for diagnosis confirmation. The patients received an intravesical instillation of BoNT-A 200 U in 30 mL normal saline after emptying their bladders and then underwent LESW treatment. The shock wave probe was placed on the transmission gel over the suprapubic region above the urinary bladder, with 3000 shocks (three pulses/s and maximum energy flow density 0.25 mJ/mm) delivered in one treatment session [16]. The patients were asked to hold their urine for at least 1 h, and the treatment was administered once a week for a total of 4 weeks. The treatment outcome was evaluated at 1 month after the fourth LESW plus BoNT-A procedure and then patients again received cystoscopic hydrodistention with bladder biopsy. The bladder specimens were analyzed with immunochemical staining for cleaved SNAP25.

All patients tolerated the treatment well without any adverse events. However, the therapeutic effect was not satisfactory. Two of the six (33%) patients had a GRA ≥ 2, and the other four (67%) had a GRA of 1. Nevertheless, immunochemical staining showed that the cleaved SNAP25 was detectable in all bladder specimens after treatment with LESW plus BoNT-A (Figure 4). The results revealed that LESW could deliver BoNT-A across the bladder urothelium and mediate BoNT-A to act on the bladder nerve endings. Although the clinical efficacy was not very satisfactory in this preliminary study, the use of LESW to promote intravesical BoNT-A delivery without the need for injection is reasonable and promising. Further study using a different LESW protocol, such as the application of LESW before BoNT-A instillation, might have the potential to improve the treatment outcome.

## 10. Future Research on the Application of LESW on Different IC/BPS Subtypes Based on the Pathophysiology and Pharmacological Mechanism

IC/BPS is a heterogeneous syndrome. Several pathomechanisms have been proposed, and the subtypes of IC/BPS might also be divided by the systemic functional somatic syndrome and other pelvic organ diseases. Treatment targeting the bladder might not be successful [98]. Bladder histopathological findings were associated with clinical parameters and differences in patient-reported treatment outcomes. Patients with IC/BPS without remarkable bladder histopathological findings had less favorable treatment outcomes compared with those who did [47]. Patients with IC/BPS can be classified into one of the following three distinct subgroups: (1) those with low glomerulation grade and MBC ≥ 760 mL, (2) those with low glomerulation grade and MBC < 760 mL or with high glomerulation grade regardless of MBC, and (3) those with Hunner’s lesions. The results showed that three IC/BPS subgroups had distinct bladder characteristics and treatment outcomes. The patients with high MBC and low glomerulation grade after hydrodistention had more medical comorbidities but a significantly higher rate of satisfactory treatment outcome [99]. Our recent research also found that Epstein–Barr virus infection in T cells might cause persistent inflammation and nerve hyperplasia in Hunner’s ulcer type IC/BPS and in patients with IC/BPS with very small bladder capacity and severe bladder pain [100].

Because of the heterogeneous composition of IC/BPS, a single treatment targeting the bladder histopathology would not adequately eradicate bladder pathology or IC symptoms. For more than 100 years, IC/BPS has remained challenging for urologists. Bladder distention alters urine antiproliferative factor activity and heparin-binding epidermal growth factor-like growth factor levels toward normal levels. Cystoscopic hydrodistention has an effect in mild IC/BPS, partly because of the stretching of the urothelium, reduction in antiproliferative factor release, and creation of an anti-inflammatory reaction [101]. Intravesical hyaluronic acid or other surface protectant instillations provide barrier protection, prevent the urine solutes from continuously stimulating the afferent nerves, and allow for the resolution of suburethelial inflammation [102]. Intravesical BoNT-A injections have strong anti-inflammatory effects in both the suburothelium and central nervous system [103]. LESW initiates new inflammation that overrides the unresolved old bladder inflammation, improves bladder perfusion, and promotes tissue regeneration [75,76]. The functional somatic syndrome involved in IC/BPS patients should also be emphasized, and psychiatric consultation helps reduce the amplification effect of mild bladder symptoms [104,105]. Therefore, clinicians have recommended the evaluation and management of IC/BPS with a multidisciplinary team to cover all possible pathogeneses of this mysterious disorder, including the use of oral pharmacotherapy, intravesical instillation or injection agents, and psychological consultation to enhance the treatment outcome [106,107]. Hence, to minimize the psychological trauma resulting from anesthesia and intravesical injections in IC/BPS patients, it is reasonable to combine some of these minimally invasive treatment modalities for patients with different IC/BPS phenotypes.

In the pilot study of LESW plus BoNT-A on IC/BPS, we only reported a preliminary result together with possible pathophysiology and a previous animal study. In the human LESW trial, the therapeutic effect of LESW monotherapy on OAB was promising [84], but the effect on inflammation of IC/BPS was limited [95], whereas LESW application plus BoNT-A instillation had been demonstrated as effective in treating IC and OAB in rats [96,97]. This evidence suggested that LESW with intravesical BoNT-A instillation might have a synergistic effect on IC/BPS. Further histological and molecular research is needed to clarify the additive effect of this combination. Nevertheless, the current therapeutic efficacy is still unsatisfactory. To achieve the best treatment outcome, the dose of LESW and BoNT-A should be further adjusted. The most suitable IC/BPS subtype should be selected on the basis of a clinical trial with a larger number of study patients. Finally, to realize the true pathophysiological changes after LESW and BoNT-A instillation, more clinical and laboratory evidence should be collected from patients who experience a therapeutic effect after LESW with or without adding BoNT-A instillation.

## 11. Conclusions

LESW has been demonstrated to induce an inflammatory reaction that might increase neovasculature, facilitate cell proliferation and differentiation, and promote wound healing in damaged tissue. Based on these effects, the use of LESW treatment is rational for improving the bladder condition of patients with IC/BPS. Animal studies have provided evidence that LESW can also increase the permeability of the bladder urothelium and promote the delivery of large-molecule proteins across the urothelium to achieve the biological effect. Since BoNT-A exerts a strong anti-inflammatory effect, the use of LESW to deliver BoNT-A into the suburothelium might increase the therapeutic effect in several inflammatory bladder disorders, especially IC/BPS. Nevertheless, current preliminary results have not been satisfactory, but with improvements in treatment technique and adjustment of the frequency of LESW and dose of BoNt-A, the therapeutic result of LESW plus BoNt-A on IC/BPS could be promising.

## Figures and Tables

**Figure 1 biomedicines-10-00396-f001:**
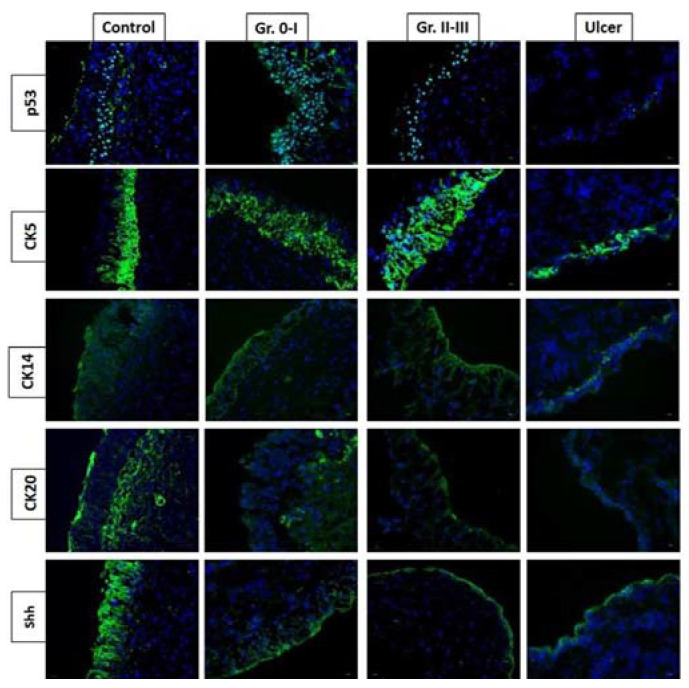
Expression of p53, CK5, CK14, CK20, and Shh proteins varies in different phenotypes of interstitial cystitis/bladder pain syndrome (IC/BPS). Hunner’s IC/BPS bladders had a low proliferative and differentiation ability, whereas the other non-Hunner’s IC/BPS bladders also had deficit proliferative and differentiation ability, both of which were also associated with the grade of glomerulations under hydrodistention.

**Figure 2 biomedicines-10-00396-f002:**
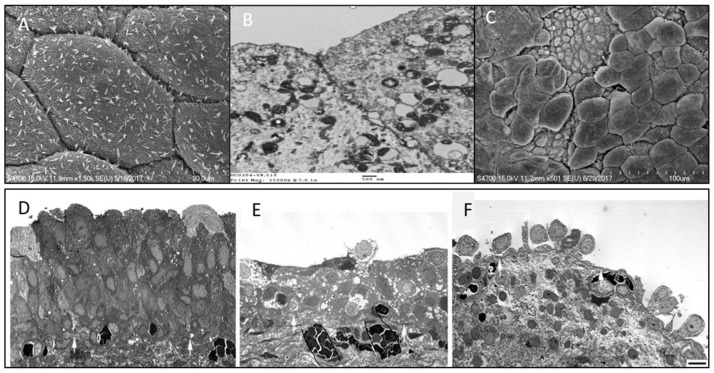
Scanning electron micrographs (EM) of the urothelium cell layer in (**A**) normal bladder urothelial cell integrity, (**B**) transmission EM showing normal tight junction of umbrella cells, (**C**) interstitial cystitis/bladder pain syndrome (IC/BPS) bladder showing defective urothelial cell layer and loss of the integrity of apical umbrella cells. Transmission EM of (**D**) normal bladder and (**E**,**F**) bladders with IC/BPS, showing different degrees of deficient urothelial cell layers and loss of the integrity of umbrella cells.

**Figure 3 biomedicines-10-00396-f003:**
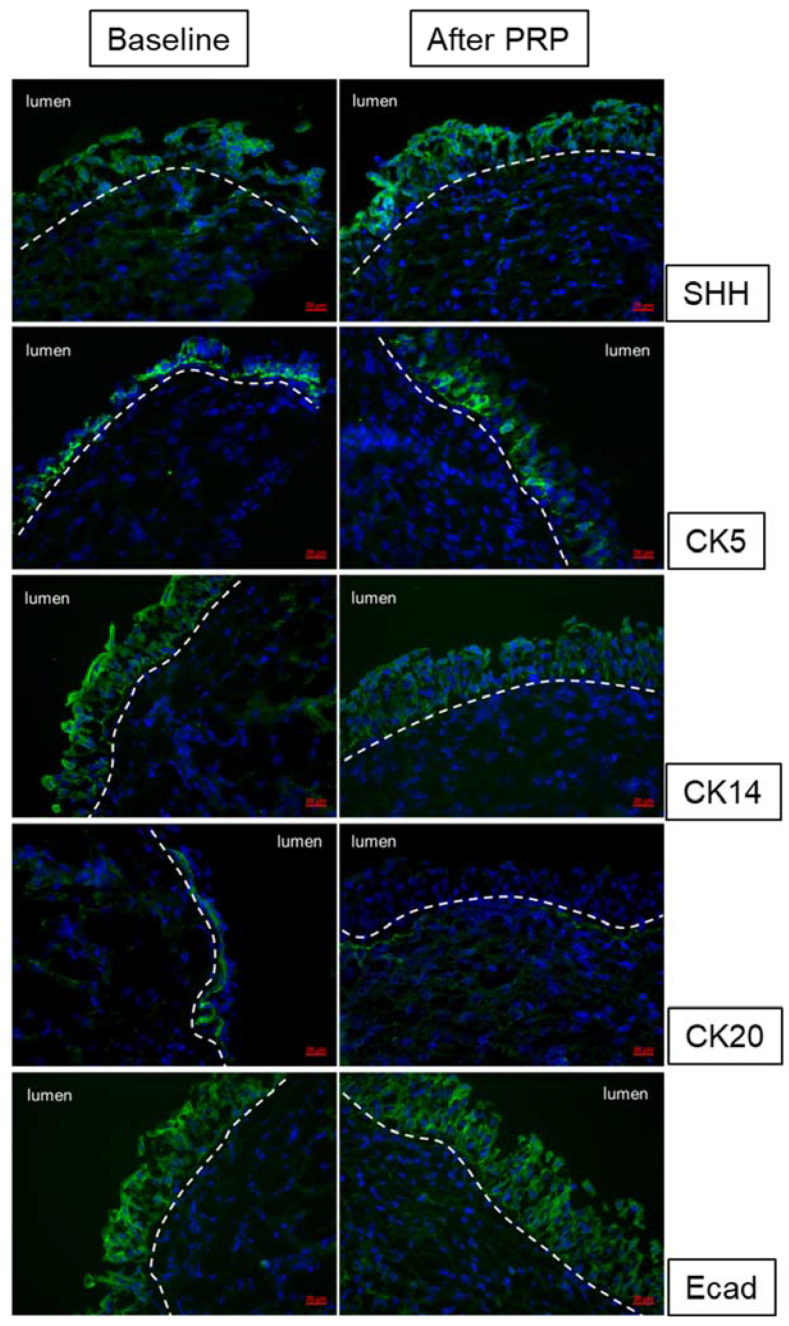
Changes in urothelium cell markers *Shh*, CK5, CK14, CK20, and E-cadherin expression after repeated platelet-rich plasma injections in interstitial cystitis/bladder pain syndrome bladders.

**Figure 4 biomedicines-10-00396-f004:**
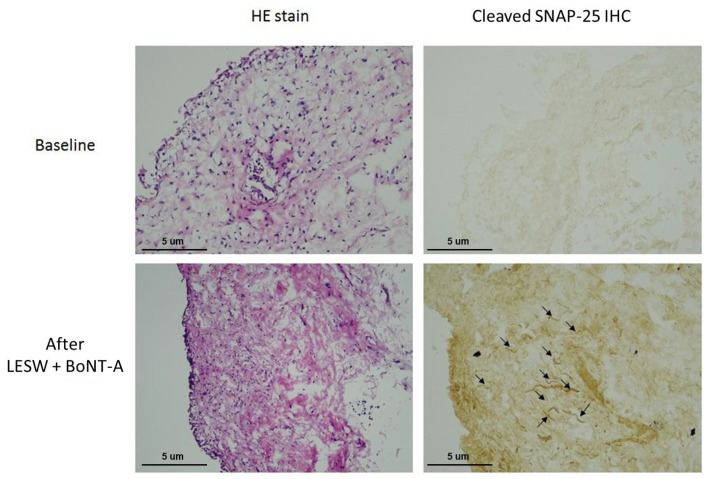
Immunochemical staining showing that the cleaved SNAP25 was not detectable in the bladder specimen before treatment (**upper**) but was evident at 1 month after LESW treatment with intravesical botulinum toxin A instillation (**lower**).

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
