# Peer review of "Low-Energy Shock Wave Plus Intravesical Instillation of Botulinum Toxin A for Interstitial Cystitis/Bladder Pain Syndrome: Pathophysiology and Preliminary Result of a Novel Minimally Invasive Treatment"

_biomedicines, 2022, doi:10.3390/biomedicines10020396_

Round 1
Reviewer 1 Report
I commend the authors for this extensive review on such important and rarely well treated disease. I found it unique and interesting. The authors extensively explored the physiopathology behind this novel treatment and its preliminary results. I only have minor advice:
- I would discuss the possible role of multi disciplinar teams in the management of this disease using these promising minimally invasive treatment
- In some points the manuscript is quite complex to read. Please try to shorten it to make it easily readable to non bladder pain syndrome experts.
Author Response
Reviewer #1
I commend the authors for this extensive review on such important and rarely well treated disease. I found it unique and interesting. The authors extensively explored the physiopathology behind this novel treatment and its preliminary results. I only have minor advice:
- I would discuss the possible role of multi disciplinar teams in the management of this disease using these promising minimally invasive treatment
- In some points the manuscript is quite complex to read. Please try to shorten it to make it easily readable to non bladder pain syndrome experts.
Reply: Thank you for the comments. We have added a statement in the last paragraph of discussion, together with the mention in this paragraph, to address the importance of multidisciplinary treatments for IC/BPS. (Lines 503-506) Further, we have shortened some statement in the biomarkers to make this manuscript more concise and easy to read.
Reviewer 2 Report
The study shows interesting data proceeding from experiments performed by the own authors as well as from several different studies directed to alleviate the symptoms of IC/BPS in human patients. There is a high number of biochemical markers analyzed in this disease and the quality of the images is very good. They concluded that LESW treatment may improve the bladder situation of patients with IC/BPS, also suggesting that a combination of LESW plus BoNT-A might increase the positive effects in different inflammatory bladder affectations.
However, in my opinion, some points should be better explained or analyzed. For instance, according to the authors report LESW induces an increased inflammatory reaction in the bladder of patients with IC/BPS, while BoNT-A exerts a strong anti-inflammatory effect, how to explain that the combination of these factors with antagonistic actions might improve the condition of these IC/BPS patients? In their study they also write about the positive effects of PRP, would it not easier to instillate GH given that this hormone induces the synthesis and release of most of factors existing in PRP? Besides it, we know that melatonin exerts a strong anti-inflammatory effect, would it be interesting to test the effect of melatonin as a liquid solution in this pathology, either alone or combined with LESW?. The study should also describe better what kind of pathologies lead to the development of IC/BPS, for instance what happens in patients that experienced chemotherapy and radiation for a bladder cancer? and what happens in patients wearing an indwelling urinary catheter? is this treatment or the combination adequate for them?
Answering to these questions or adding some of the suggestions would increase the interest and quality of this review.
There are minor typographic mistakes.
Author Response
Reviewer #2
The study shows interesting data proceeding from experiments performed by the own authors as well as from several different studies directed to alleviate the symptoms of IC/BPS in human patients. There is a high number of biochemical markers analyzed in this disease and the quality of the images is very good. They concluded that LESW treatment may improve the bladder situation of patients with IC/BPS, also suggesting that a combination of LESW plus BoNT-A might increase the positive effects in different inflammatory bladder affectations.
However, in my opinion, some points should be better explained or analyzed.
For instance, according to the authors report LESW induces an increased inflammatory reaction in the bladder of patients with IC/BPS, while BoNT-A exerts a strong anti-inflammatory effect, how to explain that the combination of these factors with antagonistic actions might improve the condition of these IC/BPS patients?
Reply: Thank you for the comments. Indeed, we only report a preliminary result together with possible pathophysiology and previous animal study. In the human LESW trial, the therapeutic effect of LESW on OAB was promising [84], but the effect on inflammation of IC/BPS was limited [95], whereas LESW application plus BoNT-A instillation had been demonstrate effective in treating IC and OAB in rats [96,97]. These evidence suggest LESW with intravesical BoNT-A instillation might have a synergistic effect on IC/BPS. Further histological and molecular researches are mandatory to clarify the additive effect of this combination. We have added this statement to the discussion section. (Lines 510-517)
In their study they also write about the positive effects of PRP, would it not easier to instillate GH given that this hormone induces the synthesis and release of most of factors existing in PRP?
Reply: Thank you for the comments. Treatment with growth factor for IC/BPS is reasonable. However, autologous PRP is easy to harvest, no allergen, and not expensive. We have used PRP injections in treatment of IC/BPS and the result was satisfactory to promote tissue regeneration and alleviate IC symptoms.
Besides it, we know that melatonin exerts a strong anti-inflammatory effect, would it be interesting to test the effect of melatonin as a liquid solution in this pathology, either alone or combined with LESW?.
Reply: Thank you for the comments. I believe melatonin will be a promising agent to combine with other treatment for IC/BPS such as adipose-derived stem cell injections. However, this topic is not in the scope of this review. (Ref. Chen YT, Chiang HJ, Chen CH, Sung PH, Lee FY, Tsai TH, Chang CL, Chen HH, Sun CK, Leu S, Chang HW, Yang CC, Yip HK. Melatonin treatment further improves adipose-derived mesenchymal stem cell therapy for acute interstitial cystitis in rat. J Pineal Res. 2014;57:248-61. )
The study should also describe better what kind of pathologies lead to the development of IC/BPS, for instance what happens in patients that experienced chemotherapy and radiation for a bladder cancer? and what happens in patients wearing an indwelling urinary catheter? is this treatment or the combination adequate for them?
Reply: Thank you for the comments. In our previous studies of IC/BPS, the inclusion criteria are in accordance with NIDDK recommendation. Patients with previous chemotherapy, radiation cystitis, bladder cancer or acute or chronic urinary retention were not included. Of course, the treatment modalities for IC/BPS can also be used in these bladder disorders.
Answering to these questions or adding some of the suggestions would increase the interest and quality of this review.
There are minor typographic mistakes.
Reply: Thank you for the comments. We have corrected the typos throughout the text.